# Urotensin 2 and Oxidative Stress Levels in Maternal Serum in Pregnancies Complicated by Intrauterine Growth Restriction

**DOI:** 10.3390/medicina55070328

**Published:** 2019-07-02

**Authors:** Esra Celik, Seyithan Taysi, Seyhun Sucu, Hasan Ulusal, Emin Sevincler, Ahmet Celik

**Affiliations:** 1Department of Medical Biochemistry, Gaziantep University Medical Faculty Hospital, 27310 Gaziantep, Turkey; 2Department of Obstetrics and Gynecology, Gaziantep University Medical Faculty Hospital, 27310 Gaziantep, Turkey; 3Balıkesir Atatürk State Hospital, 10100 Balıkesir, Turkey

**Keywords:** intrauterine growth restriction, oxidative stress, thiol-disulfide, Urotensin 2

## Abstract

*Background and objectives:* In this study, the aim was to investigate Urotensin 2 (U-II) levels and oxidant/antioxidant system parameters in pregnancies with intrauterine growth restriction (IUGR). *Materials and Methods:* A total of 36 healthy, pregnant women who had not been diagnosed with IUGR and 36 pregnant women who had been diagnosed with IUGR at the Obstetrics and Gynecology Outpatient Clinic at Gaziantep University Hospital were enrolled in this study. The serum total antioxidant status (TAS), total oxidant status (TOS), thiol-disulfide levels, U-II measurements, and oxidative stress index (OSI) calculations were carried out at the biochemistry laboratory at Gaziantep University. *Results:* According to this study, there was no statistically significant difference between the group with IUGR and the control group of healthy, pregnant women in terms of total antioxidant status (TAS), total oxidant status (TOS), oxidative stress index (OSI), native thiol, total thiol, disulfide, disulfide/native thiol, disulfide/total thiol, native thiol/total thiol, and U-II values. There was, however, a positive linear correlation between TOS and total thiol levels in the group with IUGR (*p* = 0.021, r = 0.384), and a positive linear correlation between OSI and total thiol values in the control group (*p* = 0.049, r = 0.330). In addition, there was a negative correlation between disulfide levels and gestational weeks at birth in the group with IUGR (*p* = 0.027, r = 0.369). *Conclusions:* Consequently, there was no significant difference between the control group and the group with pregnancies complicated by idiopathic IUGR in terms of serum oxidant/antioxidant system parameters and U-II levels. It is necessary to conduct more extensive studies evaluating placental, maternal, and fetal oxidative stress in conjunction in order to investigate the role of oxidative stress in IUGR.

## 1. Introduction

Intrauterine growth restriction (IUGR) is a serious complication of pregnancy and causes significant neonatal mortality and morbidity [1]. Studies have shown that poor fetomaternal circulation, genetic disorders, pregnancy-related hypertensive disorders, diabetes, perinatal infections, maternal malnutrition, exposure to toxins, and drug use are the factors that contribute to IUGR [2]. Even so, these factors account for some IUGR cases, whereas others are referred to as “idiopathic”, in which case, placental insufficiency is the most common cause of IUGR [3].

Reactive oxygen species (ROS) are highly reactive species generated by biochemical redox reactions as part of normal cell metabolism. In view of the generation of ROS, certain adverse effects also occur. The ROS and their proposed effects on biological systems have become an important area of biomedical research in recent years [4,5,6]. The increase in ROS production or the decrease in antioxidant mechanisms generates a condition described as oxidative stress, which is defined as the imbalance between pro- and antioxidants in favor of oxidants [7]. It is thought that oxidative stress plays a role in the development of placental insufficiency, which is one of the primary causes in the pathogenesis of IUGR [8].

ROS that increases in the presence of oxidative stress is the primary target of sulfur-containing amino acids of proteins. Plasma proteins, such as albumin, cysteine, glutathione, thioredoxin, and homocysteine, contain sulfur groups and are important constituents of the antioxidant defense system, wherein they play a key role in redox balance. Two cysteine residues, i.e., functional groups of these molecules, react with the electrophilic groups of ROS, become oxidized, and form disulfide bonds, and, hence, the reversible thiol-disulfide reactions are continued. The reduced state, i.e., thiol, and the oxidized state, i.e., disulfide groups, are converted in an organized manner. Consequently, the homeostasis between thiol and disulfide groups is maintained. Dynamic thiol-disulfide homeostasis is defined as a relatively new marker of oxidative stress, and according to the literature, it contributes to antioxidant defense, detoxification, and apoptosis [9].

Urotensin 2 (U-II) is a peptide that is the most potent vasoconstrictor identified to date, with receptors in the endothelium of the blood vessels, myocardium, smooth and striated muscles, adrenal gland, thyroid, and renal cortex [10,11]. It has been shown that plasma Urotensin 2 levels were increased in instances of heart failure, renal failure, diabetes, and liver disease, which had previously been found to involve oxidative and immunologic mechanisms in their etiology [10,11,12].

Antioxidant enzymes, including their activity and overall levels, and the total oxidant or antioxidant status have been studied previously [13,14]. However, serum U-II levels and markers of the antioxidant system have not previously been investigated in the same patients at the same time. The aim of this study was to investigate serum U-II, the total oxidant status (TOS), total antioxidant status (TAS), oxidative stress index (OSI), and thiol/disulfide homeostasis in the serum of patient and control groups to confirm the link between oxidative stress and IUGR.

## 2. Materials and Methods

A total of 36 control patients who were found to have normal fetal development and 36 patients who were diagnosed with intrauterine growth restriction as a result of the tests performed after they presented to the Obstetrics and Gynecology Clinic of Gaziantep University Medical Faculty Hospital between October 2017 and April 2018 were evaluated in this study. The approval dated 11.09.2017 and numbered 304 was obtained for the study from the Gaziantep University Clinical Trials Ethics Committee. After obtaining the approval of the Ethics Committee, informed consent was obtained from each patient prior to enrollment.

### 2.1. Study Population

Pregnant women with idiopathic IUGR included in the study group were aged between 18 and 35 years of age and were 34–40 weeks pregnant according to the initial ultrasonography and last menstruation date. This group did not have any coexisting conditions, such as pregnancy-related hypertensive disorders, premature membrane rupture, type 1 diabetes, gestational diabetes, renovascular disorders, sickle cell anemia, collagen-vascular diseases, hereditary-acquired thrombophilia, congenital anomaly, or multiple pregnancies. The control group consisted of healthy, pregnant women at a similar gestational age with normal pregnancy findings and normal fetal anatomy without any perinatal complications. Maternal age, maternal history, obstetric history, body height and weight, body mass index (BMI), and ultrasonography (USG) findings (biparietal diameter (BPD), head circumference (HC), abdominal circumference (AC), femur length (FL), estimated fetal weight and percentiles, amniotic fluid volume) were recorded for both groups during antenatal follow-up. Gestational week, weight, and height at birth, gender, and APGAR (Appearance, Pulse, Grimace, Activity, and Respiration) test were recorded during the postnatal period.

The collected samples were analyzed in the biochemistry laboratory of Gaziantep University Medical Faculty Hospital. Maternal venous blood samples were collected in order to determine serum TAS, TOS, thiol-disulfide levels, as well as for U-II measurements and OSI calculations, after an intrauterine growth restriction diagnosis had been made. The samples were centrifuged at 4000 rpm for 10 min to obtain the serum. Control samples were also subjected to the same centrifugation and serum separation steps and were then stored at −80 °C until the day of analysis.

### 2.2. Total Oxidant Status (TOS) Assay

The total oxidant status was assayed using a fully automated colorimetric method developed by Erel [15]. This method is based on the oxidation of Fe^+2^ O-dianisidine complex to ferric ion in the presence of the oxidant molecules within the samples and on the formation of a colored complex with the formed ferric ions and xylenol orange in an acidic environment. The color intensity associated with the amount of the oxidants in the sample is measured spectrophotometrically. TOS measurements were carried out using a fully automated TOS kit (Rel Assay DC, Gaziantep, Turkey) in an autoanalyzer (Beckman Coulter Chemistry Analyzer AU480, Brea, CA, USA). Results were expressed in µmol H_2_O_2_ Eq/L.

### 2.3. Total Antioxidant Status (TAS) Assay

The total antioxidant status was assayed using a fully automated method developed by Özcan Erel [16]. The Fe^+2^O-dianisidine complex and hydrogen peroxide (H_2_O_2_) react in a Fenton-type reaction and yield the hydroxyl radical. This potent reactive oxygen type reacts with the reducing O-dianisidine molecule that is colorless and yields the yellow-brown dianisidyl radicals at a low pH. Dianisidyl radicals take part in further oxidation reactions, thereby increasing color formation. However, antioxidants in the samples suppress these oxidation reactions and stop the aforementioned color formation. The results are obtained by spectrophotometric measurement of TAS levels in an automated analyzer following this reaction. The TAS measurement was carried out using a fully automated TAS kit (Rel Assay DC, Gaziantep, Turkey) in a Beckman Chemistry Coulter AU480 autoanalyzer. Results were expressed in µmol Trolox Eq/L.

### 2.4. Calculation of Oxidative Stress Index (OSI)

In order to calculate the OSI from the samples, the TAS unit was converted to µmol/L, and the TOS was divided by TAS to calculate the OSI [17].

### 2.5. Serum Urotensin 2 (U-II) Measurement

Urotensin 2 was measured using the competitive enzyme-linked immunosorbent assay product code: E3108Hu; Rel Assay DC, Gaziantep, Turkey. The microplate had been previously covered with human U-II antibodies. ELISA was carried out in the laboratory of the Biochemistry Department of the Gaziantep University Medical Faculty.

### 2.6. Serum Thiol-Disulfide Measurement

The homeostasis of serum thiol-disulfide was measured using a fully automated method described by Erel and Neselioğlu [18]. The measurements were carried out using fully automated native thiol, total thiol kits (Rel Assay DC, Gaziantep, Turkey) in a Beckman Chemistry Coulter AU480 autoanalyzer. This method is based on the borohydride-mediated reduction of dynamic disulfide bonds to functional thiol groups. Reducible disulfide bonds are first reduced to functional thiol groups in free form using sodium borohydride (NaBH_4_). Excess NaBH_4_ is consumed via formaldehyde and is removed from the medium. All thiol groups, the reduced ones and native thiols are reacted with 5,5′-dithiobis-2-nitrobenzoic (DTNB) acid to undertake the measurements. Half of the difference between the total thiol and native thiol yields the amount of dynamic disulfide. After the native thiol, total thiol and disulfide concentrations were detected; disulfide/native thiol, disulfide/total thiol, and native thiol/total thiol ratios were computed. The results were expressed as µmol/L.

### 2.7. Statistical Evaluation

The normal distribution of numerical variables was analyzed using the Shapiro-Wilk test. The Mann-Whitney U Test was used to compare the numerical variables that did not have a normal distribution in the two groups. The relationship between the categorical variables was determined using the Chi-Square test. The correlations between numerical variables were tested using Spearman’s rank correlation coefficient. Descriptive statistics were expressed as mean and standard deviation for numerical variables and in numbers and percentages for categorical variables. SPSS version 22.0 (IBM Corp., Released 2013, Armonk, NY, USA) was used in the analyses. *p* < 0.05 was considered statistically significant.

## 3. Results

The clinical characteristics and perinatal findings of the patient and control groups are expressed as mean and standard deviation (SD) values, and differences between the two groups are expressed as *p* values in Table 1. There was no significant difference in terms of age, gravidity, abortus, BMI (body mass indices of pregnancies), and neonatal Apgar scores at minutes 1 and 5, whereas there was a significant difference in terms of gestational week at birth, newborn weight and height, neonatal blood gas pH values, newborn hemoglobin values, and amniotic fluid indices between the two groups (*p* < 0.05). The significant results had already been anticipated having regard to the clinical course of IUGR.

Also, there was no statistically significant difference between the group with IUGR and the control group of healthy, pregnant women in terms of total antioxidant status (TAS), total oxidant status (TOS), oxidative stress index (OSI), native thiol, total thiol, disulfide, disulfide/native thiol, disulfide/total thiol, native thiol/total thiol, and U-II values (Table 2).

In addition, the correlation between the TAS, TOS, OSI, native thiol, total thiol, disulfide, and U-II levels of the groups was analyzed. Gestational week at birth and newborn weight were included in the correlation (Table 3 and Table 4). There was, however, a positive linear correlation between TOS and total thiol levels in the group with IUGR (*p* = 0.021, r = 0.384), and a positive linear correlation between OSI and total thiol values in the control group (*p* = 0.049, r = 0.330). In addition, there was a negative correlation between disulfide levels and gestational weeks at birth in the group with IUGR (*p* = 0.027, r = 0.369). Also, there was a positive correlation between the total thiol level and disulfide levels in the control group.

## 4. Discussion

Although the number of studies investigating the role of oxidative stress in pregnancies with intrauterine growth restriction is limited, earlier studies in the literature have shown that oxidative stress plays a key role in the development of placenta-related conditions [19,20]. In a study, Longini et al. [21] compared the level of isoprostane, a marker of free radical-mediated lipid peroxidation, in the amniotic fluid of pregnant women with IUGR and healthy, pregnant women, wherein it was reported that isoprostane levels were distinctly elevated in the amniotic fluid samples obtained from pregnant women with IUGR. In another study comparing pregnant women with IUGR and a control group, it was stated that the level of malondialdehyde (MDA), which is a marker of lipid peroxidation, and the activities of xanthine oxidase (XO), which is an oxidant enzyme, and glutathione peroxidase (GSH-Px), which is one of the antioxidant enzymes, were significantly increased both in maternal circulation and placental tissue [22]. Another study, including pregnant women with IUGR and preeclampsia, has shown that TAS, TOS, and OSI values were significantly increased in the group with IUGR and preeclampsia, as compared to the control group [18].

Burton et al. [8] reviewed the results of previous studies and reported that placental malperfusion was secondary to a deficit in maternal spiral artery transformation in idiopathic IUGR cases, and IUGR cases with preeclampsia play a key role in the etiology of both conditions. It was investigated as to whether different clinical signs observed in these two conditions stemmed from different underlying placental pathologies, or different maternal responses to the same placental pathology, and reported that inhibition of protein synthesis secondary to endoplasmic reticulum stress provided an explanation for the small placental phenotypes observed in both conditions; however, other pathways that are activated by more severe endoplasmic reticulum stress were only observed in the placentas of pregnancies associated with early-onset preeclampsia. When the literature was reviewed in that research, it was discovered that there is evidence supportive of the fact that the placentas in pregnancies associated with early-onset preeclampsia displayed more severe vascular dysfunction and that placental pathology was mainly focused on endoplasmic reticulum stress in pregnancies with normotensive IUGR, and, in addition, more severe oxidative stress was present in the placentas of pregnancies with IUGR + preeclampsia. The research indicated that this condition in pregnancies with IUGR + preeclampsia could lead to the release of antiangiogenic factors and trophoblastic aponecrotic debris into the maternal circulation, and to the release of potent proinflammatory cytokines that could cause a peripheral syndrome, which is observed in preeclampsia. That research also speculated that maternal and fetal constitutional factors also have a role to play in how the placenta responds to the maternal vascular damage or how the mother is affected by the placental factors released as a result of such a condition, and stated that the difference between these two conditions, i.e., IUGR and IUGR + preeclampsia, lies in the severity of the initiating deficit in spiral arterial transformation, and the degrees of endoplasmic reticulum stress and oxidative stress induced in the placenta as a result. In this current study, a comparison of the oxidant-antioxidant parameters between pregnant women with IUGR and healthy, pregnant women revealed that there was no statistically significant difference between TAS, TOS, and OSI values.

According to the current research, there are very few studies investigating the thiol-disulfide homeostasis that reflects the oxidative status in pregnancies complicated by IUGR. In this study, the researchers aimed to evaluate the thiol-disulfide homeostasis, the TOS and OSI values that reflect the oxidative status, and the TAS that indicates the antioxidant level in conjunction in pregnancies complicated by IUGR. According to this present study, there was no statistically significant difference between the group with IUGR and the control group of healthy, pregnant women in terms of total thiol, native thiol, disulfide, disulfide/native thiol, disulfide/total thiol, and native thiol/total thiol values. Cetin et al. [14] compared the thiol-disulfide homeostasis between pregnancies with IUGR and a control group of healthy, pregnant women in their study and reported that total thiol and native thiol levels were low, but disulfide levels were the same as compared to the control group. Cakar et al. [13] conducted a study to compare the total thiol, native thiol, disulfide, native thiol/total thiol, disulfide/native thiol, and disulfide/total thiol values in pregnancies with IUGR with a group of healthy, pregnant women. They divided the group of pregnant women with IUGR into two subgroups, i.e., the group with idiopathic IUGR and the group with IUGR coexisting with preeclampsia. In that study, it was found that total thiol and native thiol levels were lower in the preeclamptic IUGR group, whereas disulfide levels and native thiol/total thiol, disulfide/native thiol, disulfide/total thiol ratios were the same as in the control group consisting of healthy pregnant women. They also found that there was no statistically significant difference between the idiopathic IUGR group and the healthy control group in terms of the same parameters. These findings in this study were consistent with the results of this current research.

Min et al. [23] evaluated maternal and placental oxidative stress in pregnancies in a study and found that second trimester maternal urinary DNA damage and oxidative stress marker (8 OHdG and MDA) levels were not correlated with the values measured from the placenta at the time of birth, whereas both placental oxidative stress at birth and maternal oxidative stress during mid-pregnancy were correlated with IUGR. In this present study, the thiol-disulfide homeostasis in the maternal serum only was studied, and no significant difference was found between the study group with idiopathic IUGR and the control group. There was a positive linear correlation between the TOS and total thiol levels in the group with IUGR (*p* = 0.021, r = 0.384), according to the correlation analysis between the parameters studied in the group with intrauterine growth restriction, and the control group. Similarly, there was a positive linear correlation between OSI values and total thiol levels in the control group (*p* = 0.049, r = 0.330). Considering the fact that a high total thiol level is an indicator of a systemic reduction state, this correlation can be considered as a response to increased oxidative stress. According to the correlation analysis of this present study, there was a negative correlation between disulfide levels and gestational weeks at birth in the group with IUGR (*p* = 0.027, r = 0.369). The level of disulfide is a marker that provides an opinion regarding the systemic oxidized state. This negative correlation between disulfide levels and gestational weeks at birth implies that the gestational week achievable for birth is shorter under increased oxidized state conditions. In other words, it is possible to consider that birth will occur earlier in the presence of increased oxidative stress as compared to those at a more systemically reductive state. Oxidative stress can be commonly encountered in a normal pregnancy. However, continuous and dominant oxidative stress will lead to a decrease in and consumption of antioxidants, thereby affecting the placental antioxidant capacity and the reductive systems. Accumulating oxidative stress will cause damage to DNA, proteins, and lipids in placental tissue, thus leading to accelerated placental aging. The early aging of the placenta would prevent the placenta from satisfying the needs of the fetus and may lead to placental insufficiency and, in turn, may endanger the viability of the fetus [24]. This could necessitate the delivery of the baby prematurely in order to save the fetus. This condition can explain the shorter gestational week at birth due to the increase in disulfide levels in the group with IUGR.

Another parameter that was evaluated in this present study is a peptide, Urotensin 2. U-II is generally accepted as the most potent vasoconstrictor identified to date. U-II stimulates phospholipase C activation and Ca release induced by inositol-1,4,5-triphosphate from the endoplasmic reticulum when it binds to its receptor [25]. Previous studies have shown increased oxidative stress due to the increased endoplasmic reticulum stress (resulting in a release of reactive oxygen species and Ca^2+^ from the endoplasmic reticulum) in placental villi in pregnancies with IUGR [8,26]. In this present study, researchers investigated U-II levels in the group of pregnancies complicated by IUGR and the control group of healthy, pregnant women in order to assess whether the peptide, U-II, contributes to an increase in endoplasmic reticulum stress and oxidative stress that may be encountered in pregnancies with IUGR. There was no statistically significant difference found between the U-II levels of the two groups. Previous studies have investigated maternal serum U-II levels in pregnancies complicated by preeclampsia and have found elevated U-II levels, in comparison to a control group of healthy, pregnant women [27,28]. The patient group in this current research study consisted of pregnant women with idiopathic IUGR. Those primarily with hypertensive disorders of pregnancy, diabetes mellitus, chronic hypertension, collagen tissue disease, inflammatory diseases, asthma, and such disorders were excluded from the study. Evaluating the results from this perspective, it is thought that maternal serum U-II levels in pregnant women with idiopathic IUGR who do not have any other complications, particularly hypertension, can be similar to those in healthy, pregnant women.

## 5. Conclusions

In conclusion, in the current study, no significant difference was found in terms of serum oxidant/antioxidant system parameters and U-II levels between the control group and the pregnancy group complicated with idiopathic IUGR. The small number of patients was one of the limitations of this study. It is necessary to conduct further studies with larger patient populations in order to investigate the role of U-II and thiol/disulfide profiles in the pathophysiology of the disease in pregnancies with idiopathic IUGR.

## Figures and Tables

**Table 1 medicina-55-00328-t001:** Clinical characteristics of the groups included in the study.

Variables	IUGR (*n =* 36) (Mean ± SD)	Control (*n =* 36) (Mean ± SD)	*p* Values
Age	26.75 ± 6.20	29.19 ± 5.77	0.067
Gestational week at birth (days)	256.16 ± 17.69	265.97 ± 7.51	0.045
Gravidity	3.27 ± 2.15	4.33 ± 2.77	0.070
Abortus	0.61 ± 1.02	0.83 ± 1.84	0.777
BMI (Body mass indices of pregnancies)	28.41 ± 5.61	31.05 ± 5.65	0.149
Newborn height (cm)	42.91 ± 4.45	49.58 ± 1.18	0.001
Newborn weight (g)	2073.61 ± 525.06	3278.33 ± 349.32	0.001
APGAR 1	6.86 ± 1.53	7.41 ± 0.84	0.060
APGAR 5	8.36 ± 1.26	8.69 ± 0.70	0.115
Neonatal blood gas pH values (pH)	7.30 ± 0.08	7.34 ± 0.06	0.026
Newborn hemoglobin values (g/dL)	12.37 ± 1.38	11.62 ± 1.39	0.012
Amniotic fluid indices (AFI)	6.33 ± 5.13	9.41 ± 2.47	0.001

The results, obtained in mean and standard deviation values, were compared between the two groups. *p* < 0.05 was considered statistically significant. IUGR: intrauterine growth restriction; APGAR: Appearance, Pulse, Grimace, Activity, and Respiration.

**Table 2 medicina-55-00328-t002:** Total antioxidant status (TAS), total oxidant status (TOS), oxidative stress index (OSI), native thiol, total thiol, disulfide, disulfide/native thiol, disulfide/total thiol, native thiol/total thiol, and Urotensin 2 (U-II) values of the groups.

Parameters	IUGR (*n =* 36) (Mean ± SD)	Control (*n =* 36) (Mean ± SD)	*p* Values
TAS (µmol Trolox Eq/L)	1.76 ± 0.25	1.77 ± 0.33	0.447
TOS (µmol H_2_O_2_ Eq/L)	4.46 ± 3.93	4.13 ± 1.51	0.359
OSI (Arbitrary Unit)	0.26 ± 0.23	0.35 ± 0.36	0.100
Native thiol (µmol/L)	245.17 ± 49.95	255.00 ± 43.30	0.375
Total thiol (µmol/L)	278.00 ± 51.92	288.45 ± 45.94	0.369
Disulfide (µmol/L)	16.41 ± 5.67	16.72 ± 4.52	0.797
Disulfide/Native thiol (%)	6.95 ± 2.79	6.71 ± 2.20	0.686
Disulfide/Total thiol (%)	6.00 ± 2.10	5.85 ± 1.67	0.738
Native thiol/Total thiol (%)	87.98 ± 4.21	88.28 ± 3.35	0.738
Urotensin-II (ng/mL)	2.70 ± 1.29	2.76 ± 0.86	0.230

The results, obtained in mean and standard deviation values, were compared between the two groups. *p* < 0.05 was considered statistically significant. IUGR: intrauterine growth restriction.

**Table 3 medicina-55-00328-t003:** Correlation analysis in the group with intrauterine growth restriction (IUGR).

		TOS	OSI	Native Thiol	Total Thiol	Disulfide	U-II	Gestational Week	Newborn Weight
**TAS**	r	−0.130	−0.461	0.025	0.047	0.161	0.022	−0.209	−0.286
	*p*	0.450	0.005	0.887	0.785	0.349	0.897	0.220	0.091
	*n*	36	36	36	36	36	36	36	36
**TOS**	r		0.895	0.326	0.384	0.164	−0.108	0.058	0.133
	*p*		0.000	0.053	0.021	0.338	0.529	0.737	0.440
	*n*		36	36	36	36	36	36	36
**OSI**	r			0.197	0.259	0.140	−0.080	0.116	0.195
	*p*			0.249	0.128	0.416	0.641	0.499	0.255
	*n*			36	36	36	36	36	36
**Native thiol**	r				0.947	−0.016	−0.096	0.161	0.221
	*p*				0.000	0.924	0.576	0.348	0.195
	*n*				36	36	36	36	36
**Total thiol**	r					0.264	−0.108	0.097	0.151
	*p*					0.120	0.529	0.574	0.380
	*n*					36	36	36	36
**Disulfide**	r						0.040	−0.369	−0.282
	*p*						0.818	0.027	0.096
	*n*						36	36	36
**U-II**	r							−0.224	−0.160
	*p*							0.189	0.353
	*n*								36
**Gestational week**	r								0.779
	*p*								0.000
	*n*								36

Total antioxidant status (TAS), total oxidant status (TOS), oxidative stress index (OSI), urotensin 2 (U-II).

**Table 4 medicina-55-00328-t004:** Correlation analysis in the control group.

Control Group		TOS	OSI	Native Thiol	Total Thiol	Disulfide	U-II	Gestational Week	Newborn Weight
**TAS**	r	−0.281	−0.555	0.012	0.004	0.194	−0.001	−0.014	0.160
	*p*	0.097	0.000	0.944	0.983	0.257	0.995	0.935	0.352
	*n*	36	36	36	36	36	36	36	36
**TOS**	r		0.786	0.240	0.277	0.240	−0.028	−0.014	0.036
	*p*		0.000	0.159	0.101	0.159	0.873	0.933	0.834
	*n*		36	36	36	36	36	36	36
**OSI**	r			0.311	0.330	0.095	0.144	−0.179	−0.130
	*p*			0.065	0.049	0.582	0.404	0.297	0.449
	*n*			36	36	36	36	36	36
**Native thiol**	r				0.983	0.224	0.219	−0.207	−0.182
	*p*				0.000	0.189	0.200	0.226	0.287
	*n*				36	36	36	36	36
**Total thiol**	r					0.341	0.239	−0.159	−0.120
	*p*					0.042	0.161	0.354	0.486
	*n*					36	36	36	36
**Disulfide**	r						0.137	0.244	0.135
	*p*						0.426	0.152	0.431
	*n*						36	36	36
**U-II**	r							−0.307	−0.282
	*p*							0.068	0.096
	*n*							36	36
**Gestational week**	r								0.461
	*p*								0.005
	*n*								36

Total antioxidant status (TAS), total oxidant status (TOS), oxidative stress index (OSI), urotensin 2 (U-II).

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
