# Peer review of "Urotensin 2 and Oxidative Stress Levels in Maternal Serum in Pregnancies Complicated by Intrauterine Growth Restriction"

_medicina, 2019, doi:10.3390/medicina55070328_

Round 1
Reviewer 1 Report
This study examines oxidative stress indicators and urotensin 2 levels in women with pregnancies complicated by intrauterine growth restriction (IUGR). The study adds valuable evidence to our developing understanding of this condition.
Abstract:
Line 12 - IUGR should be defined before its first use
Line 16- OSI should be defined here rather than in line 20
Introduction:
In general, the Introduction is rather short and provides the minimal background information. It may be worth moving some of the background information from the Discussion section into the Introduction.
Line 41-42: Wording is odd, consider revising
Line 43: Somewhat colloquial
Line 56: Have these been previously studied in IUGR?
Line 57: Please also clarify whether these were previously studied in IUGR
Materials and Methods:
The TOS and TAS kits should have their supplying companies identified along with the location. The AU480 autoanalyzer should also have this information provided.
Line 113: What company were the U-II antibodies purchased from? What is their product code or clone number?
Line 119: These kits should also have their suppliers identified.
Results:
There also appears to be a positive correlation between the total thiol level and disulfide levels in the control group. This could also be described in the Results section.
Discussion:
Some areas of the Discussion are more suited to the Introduction, as they provide a lot of background information. The Discussion could be divided into more paragraphs to improve clarity.
The first long paragraph (line 167-210) provides a lot of description of previous studies. There is some unnecessary description of the details of these studies, instead the important/relevant findings of these could be summarised and referenced.
Line 172-173: The terms SGA, AGA and CAT should be defined.
Line 258-262: The authors may be overreaching to claim that the negative correlation between disulfide levels and gestational weeks indicates that birth will be earlier with increased oxidative stress. If that is the case, why is there no correlation between the oxidative stress indicators and gestational weeks?
Line 269-270: It is not completely established that increased disulfide levels cause the shorter gestational weeks. There is just a correlation.
Line 276: Ca should have a 2+ added.
Line 290-291: Confusing sentence, please revise.
Author Response
Celik et al. Urotensin 2 and Oxidative Stress Levels in Maternal Serum in Pregnancies Complicated by Intrauterine Growth Restriction.
Answers to the referees’s comments:
First of all, I would like to thank the referees evaluating our manuscript for their positive approach and valuable suggestions.
The changes can be followed as in text in bold character.
Referee: 1
Abstract:
Line 12 - IUGR was corrected as “intrauterine growth restriction (IUGR)”.
Line 16- OSI was corrected as “oxidative stress index (OSI)”.
Introduction:
Question: In general, the Introduction is rather short and provides the minimal background information. It may be worth moving some of the background information from the Discussion section into the Introduction.
Answer:
Reactive oxygen derivatives that increase in the presence of oxidative stress are the primary targets of sulfur-containing amino acids of proteins. Plasma proteins such as albumin, cysteine, glutathione, thioredoxin and homocysteine contain sulfur groups and are important constituents of the antioxidant defense system, wherein they play a key role in redox balance. Two cysteine residues, i.e. functional groups of these molecules, react with the electrophilic groups of ROS, become oxidized and form disulfide bonds, and hence the reversible thiol-disulfide reactions are continued. The reduced state, i.e. thiol, and the oxidized state, i.e. disulfide groups, are converted in an organized manner. Therefore, the homeostasis between thiol and disulfide groups is maintained. Dynamic thiol-disulfide homeostasis is defined as a relatively new marker of oxidative stress and according to the literature, it contributes to antioxidant defense, detoxification and apoptosis (9).” was added to introduction section as suggested.
Line 41-42: “Oxygen is a critical element for the living organisms and has some advantages.” removed as suggested.
Line 43: “Basically all the essential biomolecules can undergo oxidative reactions mediated by ROS.” removed as suggested.
Line 56: Have these been previously studied in IUGR?
Line 57: Please also clarify whether these were previously studied in IUGR.
To our knowledge, there is only one article investigating the levels of thiol / disulfide
homeostasis in the serum of patients with IUGR (Cetin, O.; Karaman, E.; Boza, B.; Cim, N.;
Alisik, M.; Erel, O., et al. The maternal serum thiol/disulfide homeostasis is impaired in
pregnancies complicated by idiopathic intrauterine growth restriction. J Matern Fetal
Neonatal Med. 2018,31,607-613).
There are more than one articles investigating TAS, TOS and OSI levels.
But, we could not find any articles investigating urotensin 2 levels in these patients.
Therefore, our research is different from other studies in 2 aspects and is original.
1. The only article investigating urotensin-2 levels,
2. To evaluate the levels of urotensin-2 and TAS, TOS, OSI and thiol / disulfide homeostasis simultaneously.
Materials and Methods:
Line 113: What company were the U-II antibodies purchased from? What is their product code or clone number? This information “Elabscience, CHINA” was written incorrect. It was corrected as (product code: E3108Hu: Rel Assay DC, Gaziantep, Turkey)
Question: The TOS and TAS kits should have their supplying companies identified along with the location. The AU480 autoanalyzer should also have this information provided.
Line 119: These kits should also have their suppliers identified.
Answer: “TOS measurement was carried out using a Rel Assay Diagnostics brand fully automated TOS kit in a Beckman Coulter AU480 autoanalyzer.” was changed as “TOS measurement was carried out using fully automated TOS kit (Rel Assay DC, Gaziantep, Turkey) brand in a autoanalyzer (Beckman Coulter Chemistry Analyzer AU480, CA 92821, USA).”. Others also were corrected.
Results:
Question: There also appears to be a positive correlation between the total thiol level and disulfide levels in the control group. This could also be described in the Results section.
Answer: “Also, there was a positive correlation between the total thiol level and disulfide levels in the control group.” was added to results section as suggested (Line 160,161).
Discussion:
Question: Some areas of the Discussion are more suited to the Introduction, as they provide a lot of background information. The Discussion could be divided into more paragraphs to improve clarity.
Answer: “ROS that increase in the presence of oxidative stress are the primary targets of sulfur-containing amino acids of proteins. Plasma proteins such as albumin, cysteine, glutathione, thioredoxin and homocysteine contain sulfur groups and are important constituents of the antioxidant defense system, wherein they play a key role in redox balance. Two cysteine residues, i.e. functional groups of these molecules, react with the electrophilic groups of ROS, become oxidized and form disulfide bonds, and hence the reversible thiol-disulfide reactions are continued. The reduced state, i.e. thiol, and the oxidized state, i.e. disulfide groups, are converted in an organized manner. Therefore, the homeostasis between thiol and disulfide groups is maintained. Dynamic thiol-disulfide homeostasis is defined as a relatively new marker of oxidative stress and according to the literature, it contributes to antioxidant defense, detoxification and apoptosis[1].” was added to introduction section as suggested.
Question: The first long paragraph (line 167-210) provides a lot of description of previous studies. There is some unnecessary description of the details of these studies, instead the important/relevant findings of these could be summarised and referenced.
Answer: “In another study, levels of molecules associated with oxidative stress and redox such as 8-OHdG, 4-hydroxynonenal (4-HNE), thioredoxin and redox factor-1 (ref-1) in placental tissue were compared between pregnant women with IUGR and healthily pregnant women. This study reported distinctly elevated 8-OHdG and ref-1 values in the placenta of pregnant women with IUGR (22).” removed as suggested.
“22. Takagi Y, Nikaido T, Toki T, Kita N, Kanai M, Ashida T, Ohira S, Konishi I: Levels of oxidative stress and redox-related molecules in the placenta in preeclampsia and fetal growth restriction. Virchows Arch 2004, 444(1):49-55." removed.
Question: Line 172-173: The terms SGA, AGA and CAT should be defined.
Answer: Necessary abbreviations were made as suggested.
Question: Line 258-262: The authors may be overreaching to claim that the negative correlation between disulfide levels and gestational weeks indicates that birth will be earlier with increased oxidative stress. If that is the case, why is there no correlation between the oxidative stress indicators and gestational weeks? Line 269-270: It is not completely established that increased disulfide levels cause the shorter gestational weeks. There is just a correlation.
Answer: Line 268-270, Line 258-262: “The disulfide level gives an idea about the systemically oxidized state. The negative relationship between disulfide levels and gestational weeks in the birth of the patients suggests that the gestational week that can be reached for birth decreases under conditions in which the oxidized state increases. In other words, it may be thought that in the presence of increased oxidative stress, delivery may occur earlier than those that are systemically reduced.
Oxidative stress is a common condition in a normal pregnancy, but its continuity may affect the reduction of antioxidants, leading to predominant oxidative stress, and therefore placental antioxidant capacity and reducing systems.
Accumulated oxidative stress will damage DNA, proteins and lipids in placental tissues, which will accelerate placental aging. The premature aging of the placenta will cause placental insufficiency, which will prevent the placenta from meeting the needs of the fetus and consequently endanger the viability of the fetus.
TAS and TOS values measured in this study do not fully reflect the total oxidant / antioxidant status in living organisms. There are many oxidant / antioxidant molecules that we do not measure from living organisms. In this case it must be considered.
Question: Line 276: Ca should have a 2+ added.
Answer: Ca was corrected as “Ca2+ as suggested.
Question: Line 290-291: Confusing sentence, please revise.
Answer: Line 290-291 “it is likely to be associated with various pathophysiological mechanisms, given that IUGR is a serious complication of pregnancy.” was removed as suggested.
References have been rearranged.
The language was checked by an authorized group.
We hope that the changes made are satisfactory.
Thank you very much for your patience.
Reviewer 2 Report
the expression "poor umbilical cord" in line 35 is not scientifically clear
Author Response
First of all, I would like to thank the referees evaluating our manuscript for their positive approach and valuable suggestions.
The changes can be followed as in text in bold character.
Line 35: the expression "poor umbilical cord" in line 35 is not scientifically clear.
This statement “poor umbilical cord" has been removed.
Reviewer 3 Report
This study investigate the Urotensin 2 and Oxidative Stress Levels in Maternal Serum in Pregnancies Complicated by Intrauterine Growth Restriction.
- The manuscript is well organized however the language needs some review.
- the abberviations should be elborated in the very first instance not later.
- it is not clear what cretria the authors use to recruite the control and IUGR group.
Author Response
First of all, I would like to thank the referees evaluating our manuscript for their positive approach and valuable suggestions.
The changes can be followed as in text in bold character.
Question: The abberviations should be elborated in the very first instance not later.
Answer: Necessary abbreviations were made as suggested.
Question: It is not clear what cretria the authors use to recruite the control and IUGR group.
Answer: This information “This group did not have any coexisting conditions such as pregnancy-related hypertensive disorders, premature membrane rupture, type 1 diabetes, gestational diabetes, renovascular disorders, sickle cell anemia, collagen-vascular diseases, hereditary-acquired thrombophilia, congenital anomaly or multiple pregnancies. The control group consisted of healthily pregnant women at a similar gestational age with normal pregnancy findings and normal fetal anatomy without any perinatal complications.” is available in the study population section of the material method section.